# How ‘Protein-Docking’ Translates into the New Emerging Field of Docking Small Molecules to Nucleic Acids?

**DOI:** 10.3390/molecules25122749

**Published:** 2020-06-13

**Authors:** Francesca Tessaro, Leonardo Scapozza

**Affiliations:** 1Pharmaceutical Biochemistry, School of Pharmaceutical Sciences, University of Geneva CMU, Rue Michel-Servet 1, 1211 Geneva 4, Switzerland; francesca.tessaro@unige.ch; 2Institute of Pharmaceutical Sciences of Western Switzerland, University of Geneva, 1211 Geneva, Switzerland

**Keywords:** molecular docking, nucleic acids, structure-based computational methods

## Abstract

In this review, we retraced the ‘40-year evolution’ of molecular docking algorithms. Over the course of the years, their development allowed to progress from the so-called ‘rigid-docking’ searching methods to the more sophisticated ‘semi-flexible’ and ‘flexible docking’ algorithms. Together with the advancement of computing architecture and power, molecular docking’s applications also exponentially increased, from a single-ligand binding calculation to large screening and polypharmacology profiles. Recently targeting nucleic acids with small molecules has emerged as a valuable therapeutic strategy especially for cancer treatment, along with bacterial and viral infections. For example, therapeutic intervention at the mRNA level allows to overcome the problematic of undruggable proteins without modifying the genome. Despite the promising therapeutic potential of nucleic acids, molecular docking programs have been optimized mostly for proteins. Here, we have analyzed literature data on nucleic acid to benchmark some of the widely used docking programs. Finally, the comparison between proteins and nucleic acid targets docking highlighted similarity and differences, which are intrinsically related to their chemical and structural nature.

## 1. Introduction

Nowadays computational methods are routinely applied in drug discovery campaigns and its positive impact in research is recognized in the scientific community. Computer-aided drug discovery (CADD) raised in the years 1970s and its continuous progress, together with the advancement in computer technologies, made its usage an indispensable tool. CADD methods are usually divided into two main groups: ligand-based and structure-based. The first approach builds structure-activity models based on ligand structure information, whereas the second method takes advantage of the three-dimensional structure of the biomolecular target to investigate ligand binding. Among the structure-based methods, molecular docking is a well-known tool for the identification of ligand-target recognition. In this review, we will explore the evolution of molecular docking from a single ligand-protein calculation to more recent and sophisticated applications with a particular emphasis on docking applied to nucleic acids (NA) as emerging therapeutic targets.

## 2. Nature of the Biomolecular Target for Structure-Based Methods

### 2.1. Experimentally Determined Structure

The protein data bank (PDB) (https://www.rcsb.org/) is a repository of experimental macromolecular structures, which is constantly updated. By May 2020, the PDB counted ~163′633 experimental structures of proteins, RNAs, and DNAs belonging to different organisms. The majority of the structures are solved by X-ray crystallography (~145′437) (https://www.rcsb.org/stats/summary) but also nuclear magnetic resonance (NMR) and electron microscopy (EM) techniques strongly contribute in increasing the collection. The introduction of novel protocols and methods in x-ray crystallography enormously ameliorate structure resolution if compared to the first solved myoglobin structure in 1960 [1]. Nevertheless, we are lacking structural information for many biological macromolecules. Other techniques like NMR are preferred for small systems, particularly suitable for nucleic acids as small RNA or DNA sequences where structure dynamic information might result to be crucial for the investigation of the biological event. However, Cryo-EM technology has evolved from the method for studying relatively large systems as cells organelles or macromolecular complexes but with a substantial loss on resolution, into the most recent high resolution Cryo-EM allowing to reach high resolution (better than 3 Å) for structures as small as 64 kDa [2]. This method also has the advantage of using a smaller amount of material compared to X-ray or NMR techniques.

### 2.2. Computational Methods for Structure Prediction

The PDB represents the largest source of structural data of biomolecules suitable for structure-based computational tools, and indeed for molecular docking. Despite the increasing number of deposited structures in the PDB, we are far from having structural information for all biological relevant proteins. The number of non-redundant protein sequences is estimated to be 152 million from the National Centre for Biotechnology Information (https://www.ncbi.nlm.nih.gov/refseq/). This represents an enormous gap between sequence annotation and available 3D structures. In addition, the rate at which new protein sequences are defined far exceeds the rate at which protein structures are experimentally determined. Hence, in absence of experimental structures, computational predictions emerged as a valuable alternative for structure-based studies. These computational methods can be divided into two main categories: ‘template-based’ and ‘free-modelling’. Among them, template-based methods, homology modelling in particular, results in being the most accurate for binding site identification and drug design purposes [3]. It has been estimated that with the use of homology representation, the human proteome can reach 70% of its coverage at ≥ 30% of sequence identity, which represent around 95% of the human drug targets used for therapeutic intervention [4].

Homology modelling (or comparative modelling) consists in building target models based on sequence similarity with a homologous template, which presents a known 3D structure. The general assumption of this methodology is that evolutionary related biomolecules with a similar sequence tend to have also similar structure. Normally, in the case of proteins, sequence identity with the template greater than ~30% is considered the threshold for a model suitable for structure-based studies, whereas for drug design purpose a more stringent 50% identity is required for a reliable model [5]. Sequence alignment and model validation are both crucial steps in obtaining a reliable model. Ramachandran plot, for example, geometrically evaluates backbone outliers reporting the torsion angles, ϕ and ψ, which should reside in the allowed region of the graph [1].

Homology modelling has been used for many applications in structure-based studies from target identification, site-directed mutagenesis, binding site inspection to druggability analysis. Structural models obtained by homology represents the starting point for other computational methods notably molecular docking and dynamic simulations. One example where homology models were used as a starting source of information is represented by a structure-based investigation of a novel class of aspartic proteases: *Toxoplasma gondii* Aspartyl protease 3 and *Plasmodium Falciparum* Plasmepsin IX/X, where with the application of molecular docking we could unravel the atomistic mode of action of a nanomolar inhibitor (49c) [6].

## 3. “40 Years of Protein Docking Highlights and Advancements”

After 40 years since the first application of docking, huge advancements have been made both in searching algorithms and in scoring functions, allowing its usage in a broad-range of biomolecular systems. Here, we want to highlights some of the important milestones, which contributed to the history and development of molecular docking theory and applications.

### 3.1. An Evolutionary Perspective: From Rigid to Flexible Docking

Along the course of the years, ameliorations of the docking performances mainly focused on balancing the dual aspect of predicted pose correctness vs. speed of the searching algorithm required for the calculation. The development of docking algorithms was also favored by the introduction of modern computers and parallelization, which determined upscale improvements on the speed of the calculations. Complexity of the algorithms can be classified accordingly to the number of degrees of freedom they neglect (Figure 1).

In 1982, DOCK program [7] was introduced and designed to find molecules matching by shape complementarity. Molecular and geometrical shape algorithms also called matching algorithms were firstly applied in rigid-docking calculations: where the two molecules treated as rigid bodies, explore only the six translational and rotational degrees of freedom. In the matching algorithms, the binding site is represented as a collection of spheres of different radii, which allow mapping the ligand to the sphere centers and finding matching sets [7]. Rigid docking has an appreciable speed and shows to be very effective in database exploration [8]. However, one of the major drawbacks of rigid docking is the fact of not considering ligand-flexibility and on relying only on pre-calculated conformations of the ligands. This method refers to the primordial concept of molecular recognition: the ‘lock and key’ model introduced by Fischer [9]. Because of the limited degree of freedom, virtually reducing the complexity of the system, rigid docking was successfully applied also for protein-protein and protein-peptide docking [10].

Ligand conformational freedom starts to be introduced with ligand-flexible docking methods, also referred as semi-flexible docking. Systematic algorithms, for example, generate all possible ligand conformations in order to find the best ligand-binding match. Fragmentations and incremental construction (IC) methods belong to the stochastic algorithms. With this algorithm each ligand is divided in small fragments and then rigidly docked into the binding site [11]. Only when the anchor fragment is established the rest of the molecule is joined and energy minimized in the binding site. This method works well with small to medium sized molecules, while with bigger molecules the number of generated fragments increase and thus might create difficulties in pose prediction. On the one hand, a systematic search of the ligand conformation is more accurate, on the other, it requires high computational resources and time. For instance, stochastic algorithms are a good trade-off between speed and chance to obtain the correct binding pose. Among them, Monte Carlo algorithm is a well-known stochastic method: at each iteration the ligand goes through a random bond-rotation, rigid-body translation or rotation. The binding pose is evaluated based on molecular-mechanic energy calculation and then the standard Metropolis Monte Carlo method based on the Boltzmann constant is applied to accept or reject the pose for the next iteration. A particular advantage of Monte Carlo method is the possibility to explore diverse energetic local minima conformations using higher temperature values, and therefore overcome the problem of the energy barriers [12]. Tabu method was introduced few years later with the aim of improving Monte Carlo conformational search while preventing the exploration of already sampled zones.

Genetic algorithms [13,14] also belong to the stochastic methods and they took inspiration from the biological evolution of the Darwin’s theory. Each gene represents a degree of freedom of the ligand and thus the chromosomes describe the ligand conformations. As happens in nature, mutations and crossovers occur and define new populations of conformers to which a fitness score is associated. Only the docking pose with good scoring will be selected and moved to the next iterations.

Phenomena such as allosteric modulations or certain kinetics mechanisms cannot be explained with the early ‘lock and key’ theory. Therefore, the generally accepted model regarding the molecular binding event shifted toward a more sophisticated concept of ‘induced-fit’ mechanism, more prone to explain the aforementioned events [15]. Together with these theoretical models, ‘conformational ensemble’ theory was also proposed [16], supporting the idea that different conformers exists for the same protein and that the ligand preferentially binds one of them. In this sense, the ideal molecular docking engine would allow a conformational degree of freedom on both ligand and protein. Therefore, accounting for protein flexibility is fundamental for explaining many biological processes around the ligand-binding event [17]. Molecular dynamic simulations in this sense can perfectly describe the dynamic behavior of a macromolecular system as function of time. However, the major drawback on its application is the computational cost required for analyzing in an exhaustive way the energetic landscape of the systems. Indeed, for large systems the energetic barriers that separate the possible binding modes might have results too high to be overcome, which might lead to an inadequate conformational sampling.

Before the advent of molecular dynamic, various methods tried to address the protein flexibility. For example, ‘soft docking’ [18] allows a partial overlap of the ligand-protein atoms by decreasing the van der Waals repulsion energy term employed by the scoring function. On one hand, this method results in being computationally very efficient and suitable for testing large libraries, on the other hand protein atoms are roughly fixed, approximating small movements. Just a few year later, Leach [19] introduced rotamer libraries composed by alternative conformations of amino acids side-chains. Here, the use of rotamer libraries speeds up the sampling search and allows avoiding ligand-pose minimization, which makes this approach appealing for its velocity. However, all these strategies considered a single protein conformation. Thus, large movements with the exception of side chains rotations are omitted. The ‘conformational ensemble’ docking approach recalls the homonymous theory [16], where the target exists as a collection of conformers derived by molecular dynamics or Monte Carlo simulations. Based on this theory, the ligand binding event can occur either by recognizing its preferred protein conformation or by a mutual conformational rearrangement accommodating the ligand in its binding site as induce-fit effect [15]. An alternative way to consider the ensemble of conformers has been introduced by Knegtel [20], where an energy and geometry weighted average grid is used for describing the receptor binding site. This approach has been further improved not by using an average grid but by merging the conserved moieties while creating alternative conformations for the remaining parts [21]. Today the awareness in the dynamic behavior of proteins is widely accepted and embraced by the scientific community. Sampling strategies for an exhaustive search of protein conformations have been explored during the course of the years [22,23]. Finally, the last decade of molecular docking advancements has seen the powerful integration of machine-learning methods, and even more with deep learning, which hold huge promises for computational drug discovery research [24].

### 3.2. Molecular Docking as a Crossing Tool for Multiple Scopes

From its original application molecular docking was conceived as in silico tool for investigating target-ligand interaction especially in support of drug discovery campaigns either in the screening phase, de novo design or ligand optimization. Advancements in either algorithms, scoring functions, computer architectures (GPUs), parallelization, and artificial intelligence contribute to reconfirm the crucial role of molecular docking in science (Figure 2) [25]. Indeed, it is now employed and integrated in a variety of discovery tasks not only for large library screening, but also as reverse docking for target profiling in drug repositioning, polypharmacology, and prediction of adverse effects [25,26], as shown in Figure 2.

Recent works highlight how large screening protocols involving millions of compounds will positively influence early drug discovery programs and will allow the identification of novel chemotypes belonging to their relative target [27]. The trustworthiness of molecular docking depends also on the accuracy of the used scoring function. Recent reviews have described the strengths and weaknesses of the classical scoring function as well as the more recent advancement in scoring function development [28,29]. Here we would like to highlight that artificial intelligence and statistical analysis emerged as tools to improve binding affinity and scoring functions taking advantage of the ever-growing publicly available databases [30]. One of latest applications is covalent docking or ‘reactive docking’, where results are very encouraging despite the challenging task of handling the ligand bond formation often using classical molecular mechanics force fields [31,32,33]. In this sense, new methodologies and scoring functions need to be developed to ameliorate the scoring accuracy without affecting the computational cost and hindering its application for large screening [34]. Moreover, as mentioned above, one of the major drawbacks of molecular docking is the reduced ability to handle protein flexibility. Even with the implementation of several strategies [18,19,20] it does not allow extended conformational changes of the protein structure especially at the backbone level. However, to overcome these limitations, docking is often integrated with molecular dynamics and thermodynamic calculations for both exploring the conformational ensemble and for further pose refinement [22,23,35,36]. Finally, molecular docking does not apply only to small molecules recognizing proteins but rather its broad usage allows to dock different types of macromolecules as protein-protein, peptide-protein, and not ultimately nucleic acids.

## 4. Nucleic Acids as Emerging Therapeutic Targets

Just 1.5% of the human genome encodes for proteins, only a 0.05% has been successfully drugged with existing small molecules [4]. If the latest number is considered, it appears clear that limitations on the chemical coverage within the human genome relies mainly on the difficulty to target the so-called ‘undruggable’ proteins. Disease-associated proteins represents a small fraction of the whole human genome. Therefore, if we consider that nucleic acids can be potentially modulated by small molecules then the therapeutic landscape of macromolecular targets will exponentially increase, as exemplify by the consistent fraction (70%) of the human genome transcribed into non-coding RNA [4]. DNA and RNA play essential roles in many biological processes and represent an important class of drug targets [37]. They can be distinguished by their nucleosides and sugar composition, which results into a different functional role. Indeed, while DNA is long-term storage of genetic information, RNA is responsible for the transfer of genetic information from the nucleus to the ribosome allowing protein production. From the structural point of view, DNA is composed by four nucleotide bases: adenosine, guanine, cytosine, and thymidine; while RNA presents the uracil base instead of thymidine. Another structural difference between DNA and RNA is the sugar composition: 2-deoxyribose in DNA is replaced by ribose in RNA. Each base is connected through their sugar by a phosphodiester bond, which constitutes the backbone of the DNA or RNA strand.

Small molecules interact with nucleic acids using different mechanisms: intercalation, cross-linkage, strand-cleavage, and reading-molecules. In particular, DNA-binders can interfere with the replication process, which consequently affects the transcription phase, gene expression regulation, and cell proliferation. Similarly, RNA-binders might interfere during either the transcription and translation process, which could affect the splicing machinery. Consequently, they appear as suitable targets for various diseases, not only for the chemotherapeutical area including antivirals and antibacterial [38], but they can also be exploited in genetic diseases.

### 4.1. DNA-Targeting for Cancer and Antimicrobial Therapy

DNA is considered as one of the main molecular targets in chemotherapy [39]. The discovery of first alkylating agents go back to chemical warfare during World War II, which spawned the modern era of cancer therapy [40]. Unfortunately, its non-specific interaction with the DNA cause off-target effects and high toxicity, partially limiting its use in clinic today. Also, the majority of first DNA-binders were causing double strands break which were beneficial for blocking the tumor progression, but a as drawback, this damage could lead to novel mutations and cause secondary tumors, which would appear 10–15 years later. Indeed, much more effort has been put into the development of DNA-binders without genotoxic effects. This resulted in molecules classified as groove binders, intercalators, and covalent binders. One notable example of DNA target is the oncogene c-Myc, which is over-activated in many types of cancer, the stabilization of its G-quadruplex structure inhibits the telomerase activity [41] (Figure 3A). The blockage of cell proliferation has raised interest also as therapeutic strategy for anti-bacterial treatment. Today, the potential of using DNA as a target to overcome bacterial resistance is underestimated; different strategies to increase the therapeutic index are used as the increase of the selective uptake of the drug by the microbes or designing compounds with sequence selectivity [42]. Examples of small molecules binding DNA for cancer and anti-infective therapy are reviewed in other works [39,40,42,43,44,45,46].

### 4.2. RNA as Antiviral and Antibacterial Target

The transcription and translation process can be regulated also at the RNA level and its targeting has appeared to be a suitable approach for anti-bacterial and anti-viral therapy. Riboswitches are a regulatory part of the mRNA, which control gene expression and consequently protein biosynthesis (Figure 3B). Natural ligands binding to riboswitches modulate protein gene products by upregulating or downregulating the translation process or by altering mRNA stability. The therapeutic potential of this regulatory mRNA has been widely explored as an anti-bacterial target, especially because they show high selectivity for their bound-ligand [48]. The ribosomal RNA (rRNA) is another validated target for anti-bacterial therapy, which is directly involved in the translation process. It is composed by two subunits: 30S is responsible of the correct position and reading of the tRNA and subunit 50S executing the peptide linkage. Aminoglycosides are a known class of antibiotic inducing mistranslation of the polypeptide chain and thus interfering with protein biosynthesis [49]. RNA-targeting has also raised interest. HIV trans-activation response (TAR) RNA is considered as a valid target for HIV infection and can bind macrocycle molecules that have anti-HIV activity (Figure 3C) [50].

### 4.3. mRNA Triggering Splicing Machinery

The splicing event consists in introns removal from the pre-mRNA sequence generating the mature form of the mRNA, which will then constitute the template for protein translation. The spliceosome machinery is composed by multiple proteins specifically recognizing RNA motifs. A sophisticated mechanism of interactions among the different constituents precisely rules this process [51]. Indeed, any mutations or errors during this transformation might lead to malfunction that is directly translated at a protein level (i.e., truncated proteins, misfolding, over or down expressions). Targeting splicing is an astute intervention because it occurs at the early stage of the gene expression without altering the genome [52]. Many strategies have now been developed for splicing modifications, which are used to increase specific alternative spliced isoforms or to correct aberrant gene expression caused by gene mutation altering the splicing [52]. For example, spinal muscular atrophy (SMA), a life-threatening neurodegenerative disease, is the most common genetic cause of infant mortality. Mutations at the survival motor neuron 1 (SMN1) gene directly affect the production levels of the SMN protein. A second copy of the gene, SMN2, exists, but because of a different alternative splicing pattern, it generates low SMN protein levels, which correlate with the severity of the disease. It has been demonstrated that triggering the splicing of the SMN2 gene determines exon7 (E7) inclusion and restores the lacking levels of SMN protein. Thus, this is a valid strategy for SMA treatment [53,54,55]. Recently we have reported the first target-based study of small molecule splicing modifiers binding the RNA terminal stem loop 2 (TSL2) [47]. PK4C9 was identified as a promising TSL2-binder hit, able to modify the splicing and restore SMN protein levels in SMA patient cells (Figure 3D) [47].

## 5. Current ‘Protein Docking Algorithms’ Applied to Nucleic Acids: Challenges, Solutions, and Pitfalls

Since the time of the development of molecular docking, the majority of the therapeutic targets investigated were proteins, the refinement of the docking programs was focusing mainly on these. A number of docking programs were modified and retrained in order to be applied also to nucleic acids (NA), whereas others were newly designed. Huge advances have been made to ameliorate algorithm search and scoring functions including different aspects as target flexibility, ligand geometry, and solvent effect. However, molecular docking applied to nucleic acids presents its own set of challenges and certain features might sometimes differ from the ones used for protein docking.

### 5.1. Challenges of Nucleic Acid Docking

#### 5.1.1. Structural and Active Site Features:

Similar to proteins, nucleic acids present their own set of geometrical and structural features, which characterize and distinguish them. DNA is mostly found as a double strand helix, for which three forms exists A, B, and Z. RNA can be found also as a double strand helix for which the A-form is the most present at physiological pH, whereas DNA privileges the B-form. This double helix forms two shallow cavities, the major and minor groove, which travel all along the length of the sequence. Interestingly, it has been seen that the DNA major groove is wider and accessible to large molecules as proteins and peptides. Whereas, DNA minor groove results in being narrow and more suitable to bind small molecules upon water displacements. Contrary, RNA major groove is more prone to interact with small molecules because highly electrostatic and narrower, while the RNA minor groove binds proteins via Van der Waals forces. Moreover, RNA has the characteristic to fold into different secondary structures as bulges, hairpin stem loops, G-quadruplex etc. Overall, similar to proteins, nucleic acids tertiary structures can be distinguished by specific and sophisticated geometrical motifs that form suitable drug-binding pockets able to recognized unique small molecules [56].

#### 5.1.2. Charge Distribution Effect on Ligand-Nucleic Acids Interactions:

If from a structural point of view nucleic acids results in being suitable for target-based drug discovery, the high prevalence of charges and the metal ions poses a major challenge, especially when scoring the free energy of the ligand-NA binding. This is clearly evidenced by the comparison of the electrostatic potential surface of a single nucleic acid base with the one of a neutral or charged amino acid (Figure 4).

In a solution, the negative phosphates are normally neutralized by counterions as Na^+^. Their mobile nature results indeed difficult to handle with molecular docking and often the strategy of choice is to add to the phosphate a +1 partial charge, implicitly considering the counterions [57]. The high prevalence of charges necessitates the development of dedicated scoring functions, able to describe unique ligand-NAs interactions. One of the major issues, when docking or screening positively charged ligands or amine-rich aminoglycosides is the ability to discriminate specific interactions, given the prevalence of backbone negative charges. Ribodock is an empirical scoring function developed for describing peculiar RNA-ligand interactions [58]. In particular, it accounts for a new term (S_posC-acc_), which describes a peculiar ligand-RNA or protein-RNA interaction occurring between the positive charge of a guanidine moiety (i.e., arginine residue) and the partially negatively charged carbonyl oxygen counterpart present in the nucleobases (i.e., guanine). The nature of this interaction is still debated, as it might be considered electrostatic instead of π-stacking [59].

#### 5.1.3. Metal Ions and Solvation:

The presence of the phosphate negative charges brings along a multitude of metal ions and water molecules strongly interacting to each other. Therefore, to disrupt the hydration shell ligands needs high polarity when considering the polyanionic character of the nucleic acid molecules. Site-bound ions together with waters are essential for stabilizing and shaping nucleic acids structure as shown for G-quadruplex metal-network [60] Analogous to proteins, water molecules play a major role in stabilizing the network of target-ligand interactions. Often, they are taking part in the ligand binding mechanisms. A particular challenge is represented by the water polarization effect caused by the frequent presence of charges and ions, which might reflect their effect at several layers of distances. Therefore, the role of water needs to be taken into consideration when advancing towards the ligand-optimization phase. A first step in this direction has been made in the Autodock program, where hydration is implemented as a new potential term for ‘virtually’ displaceable water molecules. This permits a substantial improvement in retrieving the native pose with RMSD values ranging below the 1.41 Å [59]. DOCK 6 introduces implicit solvation models, by means of the AMBER Generalized Born (GB) and Poisson Boltzmann (PB) in its solvation module describing the electrostatic contribution of waters and ions [61]. The combination with explicit water and counterions reaches accuracy values comparable to DrugScore^RNA^ when applied to RNA [61]. The latter is a knowledge-based, scoring function specifically designed to simulate RNA-ligand interactions. It is built on the most frequent interactions observed in crystal complexes [62]. Recently, methods like the 3D-RISM methods [60] or the SPLASH’EM scoring function [63] allowing waters and cations site-prediction during structure optimization aimed at addressing this challenging task. The accuracy of these methods still suffers from the limited number of nucleic acid-ligand solved 3D structures [63].

#### 5.1.4. Target Flexibility

As seen already for proteins, a not-negligible aspect is the intrinsic flexibility of nucleic acids structure, often ignored by docking algorithms. The binding of the ligand to the corresponding RNA or DNA is causing either an induced-fit effect [15], or a conformational stabilization/destabilization of the tertiary complex structure [16]. For example, a stabilization of the conformation of the riboswitches has been observed upon binding of the ligand [64], which results in being trapped in the binding pocket. The development of the docking programs suited for nucleic acids follows a parallel evolution to the protein docking algorithms. In reply to the target-flexibility issue, similar strategies used for protein were proposed [65]. These strategies showed substantial improvement when different cross-docking scoring were applied [66]. In this context, the MORDOR program allows flexibility on both ligand and nucleic acids by applying molecular mechanics minimization with a restrained conformational search based on the X-ray or NMR experimental structure [67]. Despite the high accuracy, this program is not recommended for screening large libraries due to the elevated computing time needed. Full flexibility of both molecules has been proposed using a Monte Carlo algorithm for ab initio ligand-DNA docking without introducing any bias on prior binding site selection [68]. Monte Carlo variables define the conformations of both ligand and DNA relative to each other and to the internal flexibility of the two molecules [68]. Another strategy to overcome the flexibility issue during the binding process is to perform docking using a conformational target ensemble that intrinsically accounts for flexibility. Our study on the PK4C9 binding to the RNA TSL2 hairpin demonstrates the importance of considering two different conformational populations, respectively tri-loop and penta-loop, where the latter preferentially accommodate the ligand [47]. Elastic potential grids have been also introduced to address target flexibility in DrugScore^RNA^ scoring function [69], as previously seen for protein-ligand docking [70]. They consisted in adapting a pre-calculated 3D grid of potential field values of an initial RNA conformation to other conformations by moving the grid intersection points in space, while keeping the potential field values constant [69]. While the use of elastic potential grids allows to consider backbone and base motion simultaneously, movements as rotational flip motions and changing in the interaction types determine the key limitations of this approach. In addition, it does not allow to model highly negative charge pockets for accommodating positively charged ligands [69].

### 5.2. Benchmark of Docking Programs Applied to Nucleic Acids

Finally, we want to rationalize the advances and performances of docking programs applied to nucleic acids along the history, which refers to works retrieved from the literature [57,71,72,73,74]. However, we have to warn the reader that literature on docking programs applied to nucleic acids is less abundant compared to protein docking and because of the small data set used, sometimes, an appropriate statistical comparison results in being difficult. Indeed, our purpose is to emphasize the evolution and the ameliorations introduced in nucleic acids docking, rather than comparing different program’s performances. A recapitulating chart is made with the intent of a figurative appreciation of the different studies (Figure 5).

The first benchmark studies date back to 2004, the original DOCK and Autodock programs were tested against a dataset of 16 RNA complexes. Both programs perform well with rigid/aromatic ligands with an average accuracy (RMSD values < 2.5 Å compared to the experimental structure) reaching the 78% pose prediction accuracy (not reported in Figure 5). Autodock outperformed DOCK when working with weak binders and aminoglycosides. Considering the overall dataset, the accuracy of the two programs ranges between 50 to 60% pose prediction accuracy [57]. In the same period, Ribodock (rDOCK in the Figure 5), a novel scoring function developed for nucleic acids, was tested in a small dataset of 10 complexes. It was able to achieve 50% of success rate in predicting the correct binding mode within 2.0 Å RMSD [58]. A slight increase in accuracy (54%) was achieved later with improvements in the solvation term of the scoring function [78]. Nine aminoglycosides were successfully docked into hydrated and flexible RNAs using Autodock with some implementation aimed at considering solvation. These results underlie the importance of including key water molecules during the calculation and the use of ensembles of conformations to assess target flexibility [59]. The MORDOR program, introduced in 2008, successfully predicted 74% of the experimental binding modes within 2.5 Å [67]. The obtained accuracy amelioration is directly linked with the improvement in handling target flexibility and the consequent induced-fit effect. Ameliorations have been implemented in DOCK version 6.0, where the use of explicit waters, sodium counterions, and rescoring with solvent models (see above) contribute to a success rate ranging from 47% to 80% depending on ligand complexity expressed by the number of ligand rotatable bonds [61]. In a comprehensive study, the performance of two widely used programs, Glide and GOLD, was evaluated over a data set of 60 RNA complexes [74]. The observed improvement in accuracy up to 65% (Glide) and 68% (GOLD) relates with the use of multiple input conformations of the ligands and of an ensemble of target conformations (specifically from NMR structures). Later studies highlight divergences in Autodock calculations between the predicted highest score geometry vs. the experimental one [72]. Indeed, combining predicted higher scores and high docking frequencies would generate better results, closer to the experimental geometries. To this purpose, a novel scoring function SPA-LN was developed for Autodock improving both prediction of binding affinity and specificity with accuracy values around 76% (a less stringent 3Å RMSD cut-off was considered) [80]. Recently, the introduction of a vibrational entropy term in the Autodock scoring function further improved the accuracy in successfully predicting the binding geometry of the ligand-DNA complex [81].

## 6. Conclusions

In this work, we have retraced through the evolution of docking programs either in terms of searching algorithms allowing the passage from rigid to flexible docking, and also in terms of applications ranging from a single-ligand docking to the prediction of a drug’s side effects, polypharmacology until nucleic acids docking. The role of this computational method has evolved, placing itself as a crossing-node tool for medicinal chemistry campaigns. Moreover, the increasing number of experimental structures deposited in the PDB and NDB databases contribute to its progress, enlarging its usage for the exploration of novel targets for new therapeutics. Despite the effort made on developing novel algorithms, ameliorating the scoring functions, docking still faces some limitations such as target flexibility. In this sense, the combination with other computational tools as molecular dynamic and sampling methods allows to overcome this limit. The advancements also in computer architectures and parallelization enable now to perform the calculation in reasonable time.

In this review, we have analyzed the use of molecular docking applied to nucleic acids, since they are emerging as valuable therapeutic targets. Many protein docking programs are also applied to nucleic acids, although, when compared to protein docking their accuracy in binding mode prediction is lower. Similar to proteins, nucleic acids flexibility, the consequent induced-fit effect, or conformational selection during the ligand-binding event are partially ignored by the docking programs. As for docking to protein, this might negatively affect the docking accuracy. Unique for nucleic acids is the electrostatic distribution of the charges, which is associated with the solvation effect of water molecules and ions distribution. New scoring functions specifically designed for nucleic acids are emerging. They include tailored terms for better describing the systems (i.e., solvation, electrostatic, entropy, etc.). Nevertheless, a major difficult task linked to the presence of metal ions and the continuous fast charge exchange between water molecules and ions remains to be solved.

In conclusion, this review reports on how molecular docking and its applications have evolved over time, the strategies adopted for partially overcoming the problems linked to ligand-macromolecule recognition, and it points out the remaining challenges to perfectly recapitulate virtually the complexity of intermolecular interactions.

## Figures and Tables

**Figure 1 molecules-25-02749-f001:**
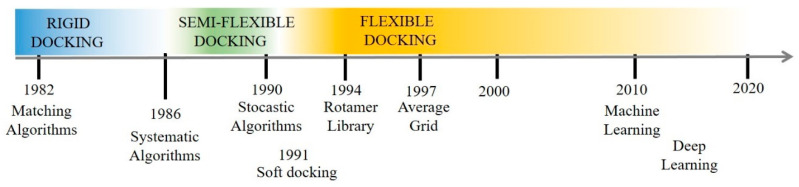
Evolutionary timeline highlighting key events contributing to the development of molecular docking algorithms, starting from rigid to flexible docking.

**Figure 2 molecules-25-02749-f002:**
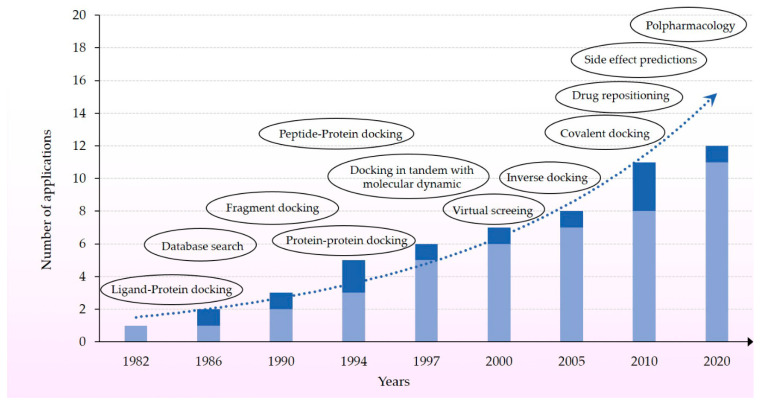
Increase of molecular docking applications. The dark blue compartment shows the new applications that are summed up with the existing ones.

**Figure 3 molecules-25-02749-f003:**
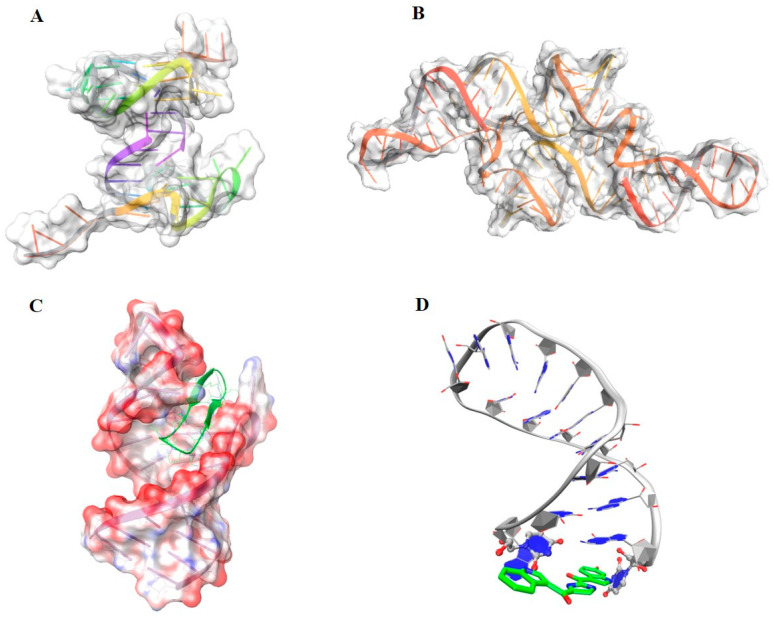
Example of nucleic acid targets. (**A**) Structure of the G-quadruplex oncogene c-Myc shown with transparent surface and backbone as cartoon style (PDB code: 6AU4). (**B**) Example of riboswitch from Glutamine II Riboswitch (PDB code: 6QN3), represented with transparent surface and the backbone as cartoon style. (**C**) Representation of the HIV trans-activation response (TAR) RNA bound to a high affinity macrocycle (PDB code: 6D2U). The electrostatic surface highlights the charge distribution, whereas the ligand bound is shown in licorice (green). (**D**) Structure of TSL2 hairpin bound with the small molecule splicing modulator PK4C9 (green). The RNA structure is shown in grey cartoon style and the ligand as licorice [47].

**Figure 4 molecules-25-02749-f004:**
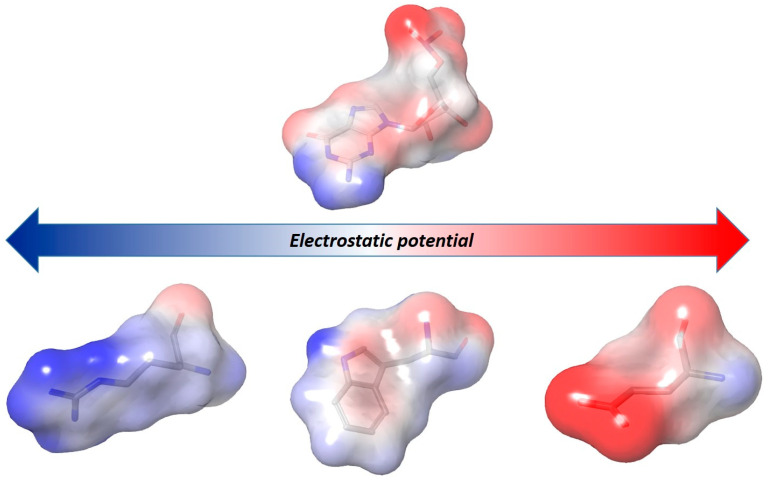
Comparison of the different charge distribution represented by electrostatic surface between polar amino acids (from left: arginine, tryptophan, and glutamate) and a nucleotide as guanosine monophosphate. The color ramp indicates in red the negative electrostatic potential energy values and in blue the positive values.

**Figure 5 molecules-25-02749-f005:**
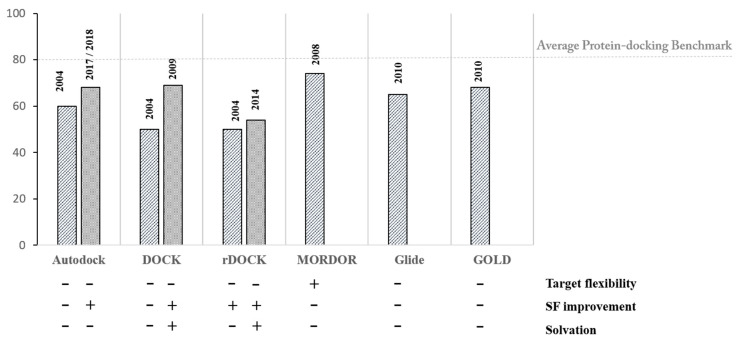
Benchmark comparison of some docking programs applied to or developed for nucleic acids. The average value of protein-docking benchmarks (80%), which has been retrieved from comparative assessments of published benchmarks [75,76,77,78,79], is used as indicator for comparison.

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
