# Peer review of "How ‘Protein-Docking’ Translates into the New Emerging Field of Docking Small Molecules to Nucleic Acids?"

_molecules, 2020, doi:10.3390/molecules25122749_

Round 1
Reviewer 1 Report
The manuscript offers an interesting review of docking methodologies and its potential application to discover NA binders. For the most part is a correct overview of the state of the field. There is room for improvement since, in my opinion, at the moment the review underdelivers a bit on its promises. There are also rather gross naming mistakes when refering to DNA/RNA structures and components that must be addressed prior to publication. Specifically:
- Sentence 49 reads weirdly: Maybe the authors meant: "Maybe authors meant "Cryo-EM has evolved"
- Sentence starting in line 55 seems a bit of a misinterpretation from the original source. Those percentages only apply under the assumption that any structure is "covered" by the PDB if it has at least 30% of sequence identity with one structure deposited in the PDB. Authors should clarify that point. That will also help reconcile their next seemingly contradictory sentence. Also on that point is unclear to me what repository are they referencing in line 59.
-
I would suggest shortening a little bit point 3. In my opinion, while the historical perspective is interesting, it has been done before, its too long (> 1/3 of the manuscript) and detracts a little bit from to the main point of the review which is the translation of protein-focused docking tools to RNA ligand discovery.
-
Authors should better frame the potential utility of targeting NA. From their current wording, if only 3.5% of the 20.000 proteins is targeted, there is potentially great room for improvement only targeting new proteins. If the authors refer to the potential undrugability of a great part of the genome, they should clarify that point and better frame the utility of DNA and RNA as potential targets beyond current therapeutical applications.
-
In line 230-231. authors should consistently refer to either the names of the nucleobases (Adenine, Guanine (NOT guanidine, which is a completely different chemical compound), cytosine, thymine and uracil) or the nucleoside names (Deoxyadenosine, Deoxyguanosine, Thymidine, Uridine, and Deoxycytidine). In fact the differences between DNA and RNA should be explained more accurately and complying with naming conventions.
-
Authors incorrectly refer to guanine (https://pubchem.ncbi.nlm.nih.gov/compound/135398634) as guanidine (https://pubchem.ncbi.nlm.nih.gov/compound/3520) throughout the manuscript
-
Please provide a reference for the statement starting in line 279: RNA-targeting has also raised interest. HIV trans-activation response (TAR) RNA is considered as a valid target for HIV intection and can bind macocycle molecules that have anti-HIV activity.
- Are the authors aware of attempts at replicating the effect of siRNA with small molecules? It would be a nice addition to the manuscript to cover these kind of approaches.
-
In figure 4 they should specify the range of electrostatic potential values in going from blue to red.
-
Authors should clarify this sentence 342-344: " In the field there is a strong debate on the nature of the peculiar interaction between the positive cabons of the guanidine and the negative carbonyl conterpart, since it might be considered electrostatic instead of pi-stacking as it was firstly described in the RiboDock scoring functions." What positively charged carbon in guanine are they refering to? Or are they refering to an arginine sidechain bearing a guanidine moiety of a protein-RNA complex?
- It would be a nice addition to the manuscript if the authors could provide a deeper discussion on the challenges of docking applied to NA and the approaches that are currently been developed to surpass them. At the moment is a little bit superficial and it accounts for a tiny part of the manuscript although it is the feature highlighted in the title. Is in this particular sense that in my opinion the review underdelivers in its own promises.
-
I am unsure of how correct figure 5 is. As the authors admit statistical comparisons are difficult due to the reduced number of studies, but moreover, different studies use diferent criteria as to what consitutes a "correct pose" raging from 2 to 3 A, which makes the differences in accuracy reported in Figure 5 artificial and definetly do not mach the y-axis label.
In any case these are easily ammendable points that should be easily corrected with minor revisions to the text.
Reviewer 2 Report
The focus of this review is on small molecules to protein/RNA/DNA docking; maybe title is bit misleading
However also protein/protein docking exists
https://www.nature.com/articles/nprot.2016.169
and protein/DNA/XNA docking:
Nucleic Acids Res. 2019 47(13):7130-7142. doi: 10.1093/nar/gkz551
typo's
175: binding
185: conceived
Reviewer 3 Report
The manuscript by Tessaro and Scapozza aims to tackle the critical and timely topic of predictions of ligand binding into the nucleic acids (NA).
Unfortunately, the content and the structure of the manuscript do not focus on this topic that much. A large fraction of the review focuses on a rather non-systematic overview of docking methods and tools, neglecting the role of scoring functions, as well as recent swarm-optimization methods, etc. Moreover, such information has already been reviewed multiple times. In this view, the review manuscript would greatly benefit from a much sharper focus on the NA docking, and as such, requires marked modification of its structure.
Other major comments:
- Title of the manuscript could be more specific to clarify its focus on docking to NA, and in fact, the focus is even more narrow (RNA only - lines 257-258).
- Section 2.2. describes protein structure prediction, which is only tangentially relevant to the topic, if the computer modeling as a source of structures is to be included, its focus should be on methods for predicting NA structures.
- Lines 80-86 discuss the use of homology models in the area of neglected diseases, without bringing any connection to the NA nor docking, and hence this text does not help to create a coherent story.
- Merging the application fields, milestones, and flexibility in Figure 1 is quite confusing. Perhaps it would be more consistent to move the application fields to Figure 2 which seems to enumerate them? Also, some milestones in Figure 1 like Machine learning and deep learning are not explained in the review at all.
- The title of Section 4 indicates that this section should be about the limitations of current docking methods when applied to NA. In this respect, Section 4.1 would much better fit Section 2 discussing biomolecular targets.
Minor comments:
- Line 66 – What is the authors’ objective to differentiate between ab initio and de novo methods?
- Lines 75-77: Example of Ramachandran plot as validation of homology modeling is a bit unfortunate as the template backbone in almost invariantly transferred to the target model.
- Lines 181-183 introduce sampling strategies for protein conformational search. It is not clear why the particular two were selected, nor what the molecular dynamics snapshots should mean, as this is clearly not a sampling strategy. Since the effective and adequate sampling is a field by itself, I would suggest just pointing to other review articles here (like refs 27, 28, or https://www.frontiersin.org/articles/10.3389/fphar.2018.00923)
- It is not clear what is meant by “reaction mechanism” on lines 329-330. Does it relate to ribozymes?
- In Figure 5, where does the value of about 80 % for the average protein-docking benchmark come from?
- I could not find Graph2 mentioned on line 380.
- There are numerous typos and grammatical errors, such as “sistematic" in Figure 1, “abondant” on line 367, or “revenge” on line 188
Round 2
Reviewer 3 Report
The revised manuscript is much improved, although I still believe that the title would benefit very much from referring directly to RNA to make the content of the manuscript clearly recognizable.